# Endothelial Function Assessment by Flow-Mediated Dilation Method: A Valuable Tool in the Evaluation of the Cardiovascular System

**DOI:** 10.3390/ijerph191811242

**Published:** 2022-09-07

**Authors:** Szymon Mućka, Martyna Miodońska, Grzegorz K. Jakubiak, Monika Starzak, Grzegorz Cieślar, Agata Stanek

**Affiliations:** 1Student Research Group, Department and Clinic of Internal Medicine, Angiology, and Physical Medicine, Faculty of Medical Sciences in Zabrze, Medical University of Silesia, Batorego 15 St., 41-902 Bytom, Poland; 2Department and Clinic of Internal Medicine, Angiology, and Physical Medicine, Faculty of Medical Sciences in Zabrze, Medical University of Silesia, Batorego 15 St., 41-902 Bytom, Poland; 3Department and Clinic of Internal Medicine, Angiology, and Physical Medicine, Specialistic Hospital No. 2 in Bytom, Batorego 15 St., 41-902 Bytom, Poland

**Keywords:** cardiovascular disease, endothelial dysfunction, flow-mediated dilation, diabetes mellitus, hypertension, dyslipidemia

## Abstract

Cardiovascular diseases (CVDs) in the course of atherosclerosis are one of the most critical public health problems in the world. Endothelial cells synthesize numerous biologically active substances involved in regulating the functions of the cardiovascular system. Endothelial dysfunction is an essential element in the pathogenesis of atherosclerosis. Thus, the assessment of endothelial function in people without overt CVD allows for a more accurate estimate of the risk of developing CVD and cardiovascular events. The assessment of endothelial function is primarily used in scientific research, and to a lesser extent in clinical practice. Among the tools for assessing endothelial function, we can distinguish biochemical and physical methods, while physical methods can be divided into invasive and non-invasive methods. Flow-mediated dilation (FMD) is based on the ultrasound assessment of changes in the diameter of the brachial artery as a result of increased blood flow. FMD is a non-invasive, safe, and repeatable test, but it must be performed by qualified and experienced medical staff. The purpose of this paper is to present the literature review results on the assessment of endothelial function using the FMD method, including its methodology, applications in clinical practice and research, limitations, and future perspectives.

## 1. Introduction

### 1.1. Cardiovascular Diseases

Cardiovascular diseases (CVDs) are among the most critical public health problems. Despite the ever-promising treatment efficacy, CVDs remain the leading cause of death in various regions of the world [1,2].

The basic process that leads to the onset of CVD is atherosclerosis, in which chronic inflammation plays a vital role in the pathogenesis [3]. The narrowing of the lumen of the artery resulting from plaque formation can lead to chronic ischemia of the supplied organ. Additionally, plaque rupture may lead to the local activation of platelet aggregation and the coagulation cascade, causing sudden restrictions of blood flow and symptoms of acute ischemia of the supplied organ [4].

The most important diseases that develop over the course of atherosclerosis include coronary heart disease (CHD), cerebrovascular disease, and peripheral arterial disease (PAD) [5]. Percutaneous transluminal angioplasty (PTA), applied optionally with stent implantation, plays a vital role in the treatment of atherosclerotic CVD. However, the development of restenosis can limit the effectiveness of treatment and contribute to the need for reintervention [6]. 

### 1.2. Endothelial Dysfunction and Its Role in the Pathogenesis of Cardiovascular Diseases

The endothelium is a single-layer epithelium that lines the inside of the blood vessels and the heart’s cavities. Endothelial well-being is a fundamental prerequisite for the proper function of the cardiovascular system [7,8].

The glycocalyx is composed of glycoproteins and proteoglycans located on the luminal surface of endothelial cells [9]. The glycocalyx, through its intracellular protein domain, which allows for endothelial cells to recognize mechanical stress, participates in the regulation of vascular wall tension, and thus in the regulation of blood distribution in peripheral tissues. The negative electric charge causes the electrostatic repulsion of blood cells from the vessel wall, which helps to maintain the fluidity of circulating blood [10]. 

Endothelial cells synthesize and secrete mediators with anti-aggregating and vasodilating properties, the most important of which are nitric oxide (NO) [11] and prostacyclin (PGI_2_) [12]. Endothelial cells also synthesize vasoconstrictive factors, including endothelin-1 (ET-1) [13] and angiotensin-converting enzyme (ACE) [14]. Endothelial dysfunction is associated with the predominant influence of vasoconstrictive, pro-thrombotic, and pro-inflammatory factors. Endothelial dysfunction is an essential element of the pathogenesis of atherosclerotic CVD [15]. 

### 1.3. Methods of the Endothelial Function Assessment

Among the different endothelial function assessments, biochemical and physical methods can be distinguished. Physical methods can be divided into invasive and non-invasive methods. The assessment of endothelial function has been widely researched and developed, and the following are examples of the most popular methods used in research.

Biochemical methods determine the concentration of selected substances in the blood, which are synthesized within endothelial cells, and their increased concentration indicates endothelial injury and dysfunction. These substances include, among others: vascular cell adhesion molecule 1 (VCAM-1), intercellular adhesion molecule 1 (ICAM-1), E-selectin, von Willebrand factor (vWF), thrombomodulin, tissue plasminogen inhibitor (t-PA), plasminogen activator inhibitor-1 (PAI-1), and metabolites of thromboxane [16,17,18,19].

Invasive physical methods consist of the intravascular administration of an agent with a vasodilating effect, depending on or independent of the function of the endothelium, and the assessment of the change in vessel diameter and blood flow using imaging diagnostics (angiography or other techniques) [20,21]. Among the noninvasive physical methods, flow-mediated dilation (FMD), enclosed zone flow-mediated dilation, digital reactive hyperemia index in peripheral arterial tonometry, venous occlusion plethysmography, and laser-based techniques can be applied [22].

More advanced methods include the assessment of glycocalyx using atomic force microscopy [23], the assessment of endothelial progenitor cells in peripheral blood [24], or the study of endothelial cells carried out in cell culture [25].

### 1.4. The Purpose of This Paper

The purpose of this paper was to conduct a review of the literature and present the most important information in the field of the current state of knowledge on the methodology of endothelial function assessment using the FMD procedure and the results of research on the use of FMD in various patient populations, particularly regarding the presence of CVD and its risk factors. Attention was also drawn to the limitations of the FMD method, and some prospects for the future are indicated.

## 2. Methodology of the FMD Procedure

### 2.1. Preparation for Examination

As a first step, the patient should be warned to avoid physical exercises, caffeine and alcohol because these factors have proven effects on short-term endothelial function [26,27,28,29]. The impact of training (at high, medium, and low intensities) was significant in the test results [30]. It is also recommended that the patient should fast for at least six hours prior to examination. The patient is forbidden to smoke twenty-four hours before the test. In addition, electronic cigarettes should be banned, even if their harmful effects on vessels are more minor than those of tobacco cigarettes [31]. The patient should be placed in a quiet room without disturbing or stressful factors. Transient stress has been shown to cause changes in the endothelium and thus falsify the FMD results [32]. Just before examination, the patient should remain in a supine position for at least ten minutes. If possible, intake of vasoactive drugs, vitamins, and supplements should be discontinued [33]. The examination is carried out in a supine position, and the patient must not make any movements.

### 2.2. Technical Background

The test involves measuring the diameter of an artery using high-resolution ultrasound and then occluding that vessel. The frequency of the waves emitted by the linear probe is 7.5–12 MHz in B-mode. Scans can be carried out manually, semiautomatically, or fully automatically [26]. A cuff that can completely close the artery is also needed. 

### 2.3. Procedure of the Examination

The aim of the examination is to measure the diameter of the brachial artery before the artery is closed with the use of a sphygmomanometer cuff (*D*1) and after the pressure is released (*D*2) [26]. From these measurements, the FMD value as a test result can be calculated according to the following equation:(1)$$FMD=\frac{D2-D1}{D1}\cdot 100\%$$

The brachial artery 3–10 cm above the ulnar fossa is the most commonly examined vessel [34,35,36]. In children, the femoral artery may also be considered. In adults, this artery has too large a diameter. On the other hand, smaller arteries are more difficult to find and make it difficult to perform a repeatable test [37]. Computer programs that help to find vessel walls can be helpful and increase the repeatability of results [38,39]. The ultrasound machine usually measures distances with an accuracy of 0.1 mm. Given that the diameter of the brachial artery is usually in the range of 3 to 4 mm, measuring the diameter to an accuracy of 0.1 mm may be highly error-prone when calculating the FMD value. Therefore, it is worth using computer software that allows for one to increase the accuracy of the measurement of the brachial diameter by one order of magnitude.

The location of the cuff appears to be of great importance for the flow-dependent vasodilation. As previously mentioned, the mechanism of arterial dilation is multifactorial after occlusion. It is suggested that distal cuff occlusion results in a largely NO-dependent dilation. This is different for occlusion proximal to the artery under test. Arterial occlusion at the proximal place may cause a dilatory response involving multiple vasoactive factors [40]. Figure 1 shows the location of the cuff in the upper arm and Figure 2 shows the location of the cuff in the forearm.

According to experts, the correct cuff pressure during arterial occlusion is 200–300 mmHg (at least 50 mmHg above the current systolic blood pressure). The relationship between the duration of occlusion and its effect on FMD has not been fully explained, but most researchers decide to occlude for five minutes [41].

A recognized disadvantage of this test is the difficulties in terms of its reproducibility, accurate standardization, and validating the obtained results [33,42]. Factors that interfere with the repeatability of the test should be considered to obtain the best results. These include the time between two measurements or the presence of hypertension [43]. Furthermore, a high reproducibility of the results is possible when experienced personnel apply standardized protocols [44]. In contrast, automatic or semi-automatic methods are acceptable but require training and standardization [35,45].

### 2.4. Reference Range

As the FMD test is mainly used in research and, to a lesser extent, in routine clinical practice, there are no clear guidelines on the range of reference values for the result of this test. Holder et al. proposed empirical formulas to calculate the mean and standard deviation of FMD values depending on gender and age, which allows for the range of reference values to be estimated depending on these two variables [46]. In a study conducted in China, in a group of healthy children and adolescents aged from eight to eighteen years, FMD values ranged from 8.29% to 8.80% in females and 8.34% to 8.77% in males [47]. Maruhashi et al. proposed a value of less than 7.1% as a cut-off point for diagnosing FMD endothelial dysfunction [48].

## 3. FMD in Patients with Obesity, Metabolic Syndrome, and Its Components

### 3.1. FMD Results in Patients with Obesity

A statistically significant negative correlation was found between the value of the waist circumference and the value of the FMD test result [49]. A lower brachial FMD was observed in patients with extreme obesity. Williams et al. and Gluszewska et al. demonstrated that weight loss after bariatric surgery was positively correlated with improved FMD. Both studies aimed to determine whether obesity can influence endothelial function in patients with abnormal FMD results before surgery; after successful intervention, FMD significantly improved in both studies [50,51]. According to Gluszewska et al., carotid intima-media thickness (IMT) also significantly decreased six months after bariatric surgery. Dominik-Karlowicz et al. conducted a similar study. They showed that, after bariatric surgery, patients arterial performance factors, such as FMD and pulse wave velocity (PWV), improved. However, a negative correlation was observed between FMD and body mass index (BMI) [52], which is also reflected in the literature [53]. 

Researchers indicate a multifactorial determination of endothelial impairment. First, visceral adipose tissue has a high bioactive potential [51]. Leptin, one of the most essential secretions by adipose tissue, appears to stimulate endothelial inflammatory reactions, making it a proatherogenic substance [54]. Interestingly, the dysfunction of perivascular adipose tissue may play a special role in the pathogenesis of CVD [55]. Oxidative stress and the increased production of inflammatory cytokines reduce NO secretion, contributing to reduced vasodilation [56].

Ne et al. in their meta-analysis, demonstrated that FMD is significantly lower and IMT substantially higher in obese patients. Gender and age did not significantly affect the relationship between obesity and FMD and IMT values. However, there was a significant heterogeneity based on the I^2^ test [56].

Wycherley et al. examined the effect of a low-carbohydrate diet in obese patients with type 2 diabetes mellitus (DM). The switch to a very low carbohydrate diet did not have a significant impact on FMD, despite the reduction in weight and the decrease in HbA1c levels [57].

### 3.2. FMD Results in Patients with Metabolic Syndrome

Metabolic syndrome (MS) is a constellation of abnormalities in energy balance [58]. One of the parameters is centrally distributed obesity, which significantly contributes to insulin resistance [59] and is directly associated with a higher risk of CVD death [60]. It is worth noting that obesity is associated with increased oxidative stress [61], and “obesity and insulin resistance” is the MS component that contributes the most to this relationship [62]. Oxidative stress plays a vital role in the development of endothelial dysfunction and pathogenesis of atherosclerosis [63,64,65]. 

Central obesity is a factor that correlates significantly with reduced FMD. Furthermore, reduced FMD in patients with MS increased the risk of cardiovascular events [49]. Ryliškytė et al. examined the arterial function and the role of the endothelium in patients with MS using FMD, IMT, aortic augmentation index (Alx), cardio-ankle vascular index (CAVI), and PWV tests. The researchers found that reducing FMD by one standard deviation increases the risk of a cardiovascular incident by 17%. In addition, depending on the IMT score, FMD testing can be used at low cardiovascular risk or for PWV measurements at higher risk. Moreover, patients without a history of CVD had significantly better FMD, IMT, PWV, and mean blood pressure [66].

FMD can also be useful in children. Research conducted in 2013, which included 38 obese children and 34 controls, showed that FMD and IMT are valuable methods when identifying patients with increased cardiovascular risk [67].

### 3.3. FMD Results in Patients with Diabetes Mellitus

Lockhart et al. studied FMD in type 1 DM under diastolic shear stress. Their results showed that FMD is significantly impaired in patients with type 1 DM compared to controls. Similarly, endothelium-independent dilatation was considerably damaged in response to glyceryl trinitrate (GTN). Pulse waveform was also different in the study and control group [68]. Hamilton et al. also concluded that type 1 DM promotes endothelial dysfunction based on abnormal FMD results, despite the imprecision of the test technique [69]. Shivalkar et al. investigated the impact of type 1 DM on cardiovascular risk using FMD and IMT assessment. The results showed a decrease in FMD and an increase in IMT in patients. Regarding the patient’s condition and laboratory tests, FMD was found to be one of the best predictors of cardiac dysfunction in patients with type 1 DM [70]. Children with type 1 DM had significantly increased IMT and decreased FMD. Additionally, elevations in markers such as leptin, tumor necrosis factor α (TNF-α), interleukin 4 (IL-4), and high-sensitivity C-reactive protein (hs-CRP) were significantly different between patients with and without diagnosed type 1 DM [71].

A significant predictor of future cardiovascular events is coronary artery calcification (CAC), measured by computed tomography. In the study by Ono et al., patients with confirmed CAC and comorbid DM had significantly elevated IMT and reduced FMD. Researchers point to the benefits of combining FMD and IMT to assess cardiovascular risk in patients with DM [72]. Barchetta et al. conducted a study to investigate whether elevated inhibitors of dipeptidyl peptidase *4* (DDP-4) activity are correlated with FMD. In a group of 62 patients with type 2 DM, plasma DDP-4 activity was higher than in a control group with the same number of healthy patients. Additionally, DDP-4 values were correlated with higher BMI, waist circumference, the blood level of transaminases, and nonalcoholic fatty liver disease (NAFLD). Finally, FMD values were markedly reduced and associated with elevated plasma DDP-4 activity. However, IMT values were negatively correlated with DDP-4 [73]. The contribution of exercise to improvements in endothelial function in patients with type 2 DM was studied by Ghardashi Afousi et al. They showed that low-volume high-intensity interval training (LV-HIIT) for twelve weeks has a significantly positive effect on FMD and caused it to increase [74]. The decrease in FMD was not only correlated with the diagnosed DM, but also with elevated fasting glucose levels. In those cases, patients additionally suffered from chronic kidney disease (CKD) [75] or MS [49]. A decrease in nitroglycerine-mediated flow dilation (NMD) and an increase in IMT were observed in patients with diabetic angiopathy [76].

### 3.4. FMD Results in Patients with Hypertension

Chronic hypertension damages the vessels by inducing inflammation and then the development of atherosclerosis [77]. Impaired FMD is highly correlated with at least moderate hypertension. This appears to be related to a reduced ability to produce endothelial nitric oxide [78]. IMT, brachial-ankle PWV, and FMD (although less significantly), mainly when performed together, were shown to be a valuable tool to predict the risk of future cardiovascular events risk in elderly patients. Moreover, male gender and hypertension were the most common risk factors for vascular complications [79].

The detection of subclinical atherosclerosis is essential to improve patient prognosis. Children of hypertensive parents belong to a particular risk group. A related study was conducted by Evrengul et al. They measured FMD in non-hypertensive offspring of hypertensive parents. They found that offspring of hypertensive parents had lower FMD results than offspring of non-hypertensive parents. The aortic stiffness test noted the same relationship [80]. Cetin et al. found an inverse correlation between FMD and left atrial minimum and maximum volumes in hypertensive patients [81].

### 3.5. FMD Results in Patients with Dyslipidemia

Rinkūnienė et al. found a significant relationship between MS, hypertriglyceridemia, and deterioration of arterial performance parameters such as lower IMT and AIx adjusted for a heartrate of 75 beats per minute, as well as higher PWV and mean arterial pressure (MAP). No significant difference was found in FMD value (*p* = 0.283) [82].

The chronic inflammatory process and endothelial dysfunction are the most critical mechanisms in the pathogenesis of atherosclerosis. Statins, the most crucial group of lipid-lowering drugs, also have anti-inflammatory properties. Therapy with a monoclonal antibody targeting interleukin-1β shows promising results and contributes to a reduction in cardiovascular incidents compared to the placebo group [83]. Patients with dyslipidemia had lower FMD and higher carotid IMT than the healthy reference group [84]. Moran et al. found a significantly higher FMD value in patients with dyslipidemia over the course of resistance to thyroid hormone β than in controls, but no significant difference was found in the IMT value. This may be associated with endothelial hyperreactivity caused by an increase in thyroid hormone levels [85].

## 4. FMD in Patients with Cardiovascular Disease

FMD was shown to play an essential role in assessing cardiovascular event risk. According to a meta-analysis by Inaba et al., the pooled multivariate relative risks of cardiovascular events per 1% and per one standard deviation increase in FMD were 0.872 (95% CI 0.832–0.914) and 0.593 (95% CI 0.490–0.718), respectively [86].

### 4.1. Coronary Artery Disease

Gupta et al. found a substantial decrease in FMD in patients after myocardial infarction compared to controls. Other parameters of vascular function, such as IMT and ankle-brachial index (ABI), were also significantly altered in this population [87]. In another study, the parameters of vascular function were assessed in patients with CHD undergoing coronary angiography. A significant decrease in FMD was observed in patients with CHD, and the downward trend continued with the increasing number of diseased coronary vessels and SYNTAX score. Moreover, as the number of diseased vessels increased, the IMT value increased. In addition, the researchers observed the presence of factors that predispose one to CHD, such as age above 69 years, male sex, arterial hypertension, DM, dyslipidemia, low FMD, and high IMT [88].

### 4.2. Peripheral Arterial Disease

In a study in which patients with PAD treated with PTA participated, Kaczmarczyk et al. compared data before the procedure, as well as one, six, and twelve months after PTA. FMD results slightly improved one month after the procedure, but after six months, they decreased to the pre-treatment values. IMT decreased after the PTA procedure compared to before the procedure. ABI first slightly increased, then maintained worse values until the end of the study. Toe-brachial index (TBI) increased after endovascular treatment and remained at one level. Despite the lack of spectacular results in terms of indicators of vascular function, researchers point to the benefits of the PTA procedure due to the clinical improvement in patients, as assessed by pain-free walking distance and maximal walking distance [89].

### 4.3. Chronic Kidney Disease

CVD is the leading cause of death in patients with CKD [90]. Changes in biochemical parameters such as ICAM-1 and asymmetric dimethylarginine (ADMA) are frequently observed in patients with CKD and may indicate endothelial damage [91]. Patients with CKD have reduced FMD. There is a significant correlation between elevated CRP values and decreased FMD in CKD patients [92].

The role of dyslipidemia in endothelial damage is not clear in patients with CKD. Bai et al. showed that in patients with CKD and MS, FMD is significantly lower compared to patients without CKD and without MS. According to a Pearson correlation analysis, FMD was significantly negatively correlated with waist circumference in women (*r* = −0.223, *p* = 0.03) and fasting blood glucose (*r* = −0.186, *p* = 0.001), while no significant correlation was found between FMD and lipid blood parameters [75]. Dogra et al. confirmed that FMD is significantly decreased in patients with CKD when compared to healthy controls. Moreover, dyslipidemia was not associated with vascular dysfunction in CKD patients. Insulin resistance and systolic blood pressure were negatively correlated with FMD in patients with CKD [93].

Verbeke et al. found that FMD is significantly decreased in patients with end-stage renal disease (ESRD) without diagnosed CVD compared to controls and is also significantly decreased in patients with coexisting ESRD and CVD [94]. In other studies, attenuated FMD was strongly correlated with proteinuria [75,95].

Impaired FMD was positively correlated with worsening renal glomerular filtration rate but was not associated with increased mortality. Deterioration of the abdominal aortic calcification score, plasma cardiac markers (troponin and natriuretic peptides), and echocardiographic parameter E/e’ contribute to increased mortality in patients with CKD [96]. Miyagi et al. confirmed a significant correlation between FMD and estimated glomerular filtration rate (*r* = 0.31, *p* = 0.0002), and a significant negative correlation was found between FMD and small artery intimal thickening (*r* = 0.54, *p* = 0.0001) [97].

After kidney transplantation, improvements in FMD were evident (from 9.1 to 15.7%, *p* < 0.001) [98].

## 5. FMD Results and Biochemical Changes

Rueda-Clausen et al. show that patients with dyslipidemia and a clinical history of CHD have higher CRP, IL-6, and sVCAM-1 blood levels compared to patients with dyslipidemia and no history of CHD. Elevation of the mentioned markers was associated with higher carotid IMT, but no significant differences were found in FMD [84].

Interesting observations were made by Bartoli et al. and Abdou et al. when studying the vascular effects of systemic sclerosis (SSc). According to Bartoli et al., these were significantly reduced in patients with SSc compared to controls (3.41 ± 4.56% vs. 7.66 ± 4.24%; *p* < 0.037), although no correlation was found between FMD and autoantibody patterns, as well as variables such as disease duration, SSc subset, capillaroscopic pattern, pulmonary involvement, and the presence of digital ulcers [99]. Abdou et al. also found that FMD is significantly decreased in patients with SSc. However, FMD is positively correlated with steroid dose (*r* = 0.385, *p* = 0.048). Contrary to IMT, FMD did not significantly correlate with CRP concentration [100]. However, according to Pacholczak-Madej et al., the decrease in FMD in SSc patients can be partially explained by the increased level of CRP in a simple regression model (*β* = −0.38, 95% CI −0.55 to −0.22) [101].

An important determinant of vascular impairment is oxidative stress. Majer et al. focused on verifying whether vessel function parameters (FMD, ABI, IMT) can correlate with antioxidant vitamin levels (A, D, E) [102]. Patients with FMD levels greater than 8.8% were found to have significantly higher plasma concentrations of ascorbic acid, retinol and α-tocopherol than those with FMD below this threshold. Furthermore, vitamin A and E levels were positively correlated with parameters of arterial wall function (FMD) and hemodynamics in lower extremity arteries (ABI), but not with pulse pressure and IMT [102]. In a meta-analysis performed by Joris and Mensink, it was confirmed that vitamin E supplementation significantly improves fasting FMD by 2.42% (95% CI 0.46% to 4.37%; *p* = 0.015) [103]. 

Supplementation with vitamin D in patients with arterial hypertension and type 2 DM was associated with a significant increase in FMD test results and a significant decrease in levels of ox-LDL and ICAM-1 after twelve weeks [104]. Vitamin D is associated with calcium metabolism, but receptors for cholecalciferol have also been identified on the surface of endothelial cells [105]. This could explain the direct effect of vitamin D on nitric oxide synthesis, which is the primary substance that dilates vessels (including the vasodilation observed in FMD) [106]. However, in the above-mentioned meta-analysis performed by Joris and Mensink, no effect of vitamin D supplementation on FMD value was found (0.15%; 95% CI −0.21% to 0.51%; *p* = 0.41) [103], so the effect of vitamin D on endothelial function remains unclear.

## 6. FMD Results and Physical Activity

Physical exercise is a factor that improves endothelial function by resting peripheral conduit artery shear profiles [107]. Undoubtedly, daily activity promotes increased NO availability and delays the atherosclerotic process. The meta-analysis performed by Cornelissen showed that physical exercise, depending on its type, has a positive effect on lowering blood pressure. The best results are visible with aerobic exercise [108]. The reasons for this mechanism are complex and not fully clear. It is assumed to be involved in compensatory sympathetic withdrawal, resetting the baroreflex, and reductions in peripheral vascular resistance [109].

The study by Mazurek et al. involved people who had no disease burden and declared daily physical activity. The aim was to determine whether gender could influence FMD and IMT results. The most significant differences were found in the IMT value, which increased more than in women. FMD results did not vary between groups [110].

Moriguchi et al. analyzed whether exercise improves FMD in people with mild hypertension. At the beginning of the study, they compared FMD after NO infusion between a study group and a control group. Vasodilatation was similar in both groups. After a twelve-week exercise period, the study group had significantly better FMD results than the non-exercise control group [78]. In other studies, exercise, including supervised training on a treadmill for twelve weeks, led to improved FMD and reduced stress markers. In addition, patients’ walking ability improved, and intermittent claudication stopped [111,112]. The type of exercise also seems to make a difference. Ghardashi Afousi and his colleagues conducted a study in which patients with type 2 DM were followed for twelve weeks for FMD, looking at the type of physical activity. They compared low-volume high-intensity interval training (LV-HIIT) and continuous moderate-intensity training (CMIT). Post hoc analysis demonstrated that the increase in FMD from baseline was higher in the LV-HIIT group than in the CMIT group [74].

## 7. FMD Results in Pregnancy

FMD testing was also used in the assessment of vessels in pregnant women. Significant FMD impairments were observed in preeclamptic (PE) and gestational hypertensive (GH) women. The situation was different in patients who were taking GH or PE medication. FMD in GH-treated women was significantly reduced. On the other hand, women treated for PE did not have a significant worsening of FMD [113]. According to Weissgerber’s meta-analysis, reduced FMD was present before and did not resolve three years after childbirth [114]. Similar conclusions were reached by researchers from India who tried to assess whether the risk of PE or GH in pregnancy could be predicted using FMD. Their observations suggest that FMD scores decrease as the pregnancy advances. The decrease is more significant in women who developed GH/PE than in healthy women. After the end of pregnancy, FMD returns to the normal range in both groups [115].

## 8. FMD Results in Patients with COVID-19

The outbreak of the COVID-19 pandemic has caused many health-related adverse effects, including diseases of many organs, vessels, and even psychiatric problems [116,117,118,119]. Ergül et al. concluded that previous COVID-19 infection and abnormal BMI are independent predictors of endothelial damage. Furthermore, the researchers point to increased cardiovascular morbidity and mortality after SARS-CoV-2 virus infection [120]. Lambadiari et al. investigated abnormal FMD and markers of oxidative stress. They found the above-mentioned abnormalities in patients four months after COVID-19 infection [118]. Riou et al. studied patients three months after COVID-19 recovery. Their finding shows that the severity of SARS-CoV-2 virus infection had no significant effect on endothelial function deterioration, although FMD was significantly lower in patients after COVID-19 compared to controls (8.2 vs. 10.3; *p* = 0.002) [121].

## 9. Current Limitations of FMD Procedure and Future Trends

According to Stoner et al., there are a few main limitations associated with the standard FMD methodology: first, inappropriate expression of FMD, second, measurement variance associated with a short-lived FMD response, third, most studies fail to account for the FMD stimulus, and lastly, poor reproducibility [122,123].

It has been shown that shear rate-diameter dose–response curves could improve the traditional FMD measurement and serve as a superior clinical and research tool for assessing cardiovascular disease risk in various populations [123].

Tremblay et al. noted that the shear forces inducing vasodilation during the classic FMD procedure described in this paper, called reactive hyperemia-induced flow-mediated dilation (RH-FMD), do not coincide with the forces that affect the endothelium in vivo. Therefore, a modification of the FMD method, called sustained stimulus-flow mediated dilation (SS-FMD), was proposed, in which the stimuli that induce vasodilation are factors such as limb-heating, distal vasodilator infusion, and exercise. At present, the RH-FMD method is the better-studied in vivo method in humans, so further studies are needed to establish the clinical utility of SS-FMD. The research results suggest that the RH-FMD and SS-FMD methods assess slightly different aspects of endothelial function [124].

## 10. Conclusions

Endothelial well-being is one of the basic conditions for the well-being of the entire cardiovascular system, and endothelial dysfunction is the first stage in the development of atherosclerosis. CVDs, especially in the course of atherosclerosis, are one of the most important causes of morbidity and mortality worldwide. To effectively treat CVDs and prevent complications, it is essential to identify patients with increased cardiovascular risk and to identify cardiovascular dysfunctions at an early, subclinical stage. One of the directions of the attempt to implement this concept is the development of methods to assess endothelial function. FMD is of great interest, although it is mainly used in research and, to a much lesser extent, in routine clinical practice. The advantages of the FMD method include its noninvasive nature and relatively low equipment requirements. Significant limitations include the need to perform the test with experienced personnel, leading to low reproducibility between different investigators and centers. In addition, care should be taken to ensure that the test is carried out in appropriate conditions.

In the opinion of the authors of this review, conducting an FMD study may be particularly valuable in the case of patients without apparent CVD who are diagnosed with modifiable cardiovascular risk factors, such as DM, dyslipidemia, and hypertension. In this case, the FMD test could be repeated after adequate treatment has been initiated. The subclinical dysfunction of the cardiovascular system should also be supplemented with additional elements, such as ABI, TBI, IMT measurements, and the assessment of arterial stiffness.

This paper presents the results of a literature review on the methodology of FMD research and the effects of scientific research using this method in various clinical situations. 

Table 1 presents the essential conclusions resulting from this review of the literature.

## Figures and Tables

**Figure 1 ijerph-19-11242-f001:**
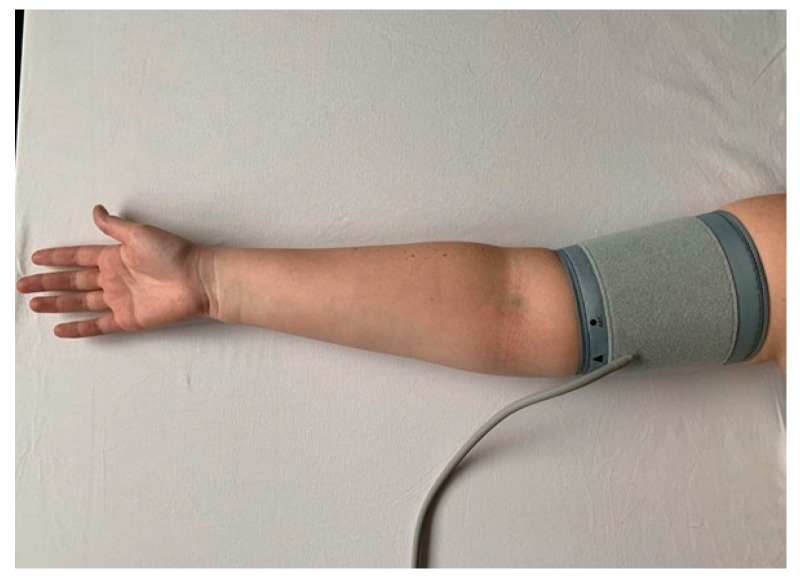
Proximal location of the cuff.

**Figure 2 ijerph-19-11242-f002:**
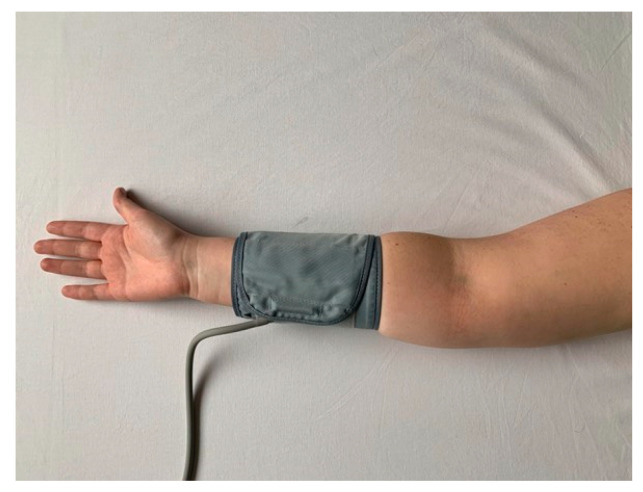
Distal location of the cuff.

**Table 1 ijerph-19-11242-t001:** The most important findings of this review of the literature.

Over the course of obesity, significantly lower FMD values are observed and, after successful bariatric treatment, a significant increase in FMD values was observed [49,50,51,52,53].
FMD is significantly associated with cardiovascular risk in patients with metabolic syndrome (MS) [66].
FMD is significantly reduced compared to controls in patients with type 1 diabetes [68,69,70,71] and in patients with type 2 diabetes [73,74,76].
The FMD test confirms endothelial dysfunction in patients with hypertension [78,79] which is associated with cardiovascular dysfunction assessed by other methods, such as intima-media thickness and pulse wave velocity [79].
The influence of dyslipidemia on the value of FMD is not fully clear [84,85].
FMD is a valuable method for assessing patients with already diagnosed cardiovascular disease [87,88] or chronic renal failure [94,95,96,97,98].

## Data Availability

Not applicable.

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
