# Peer review of "Endothelial Function Assessment by Flow-Mediated Dilation Method: A Valuable Tool in the Evaluation of the Cardiovascular System"

_ijerph, 2022, doi:10.3390/ijerph191811242_

Round 1
Reviewer 1 Report
I read with interest this review by Mucka et al.
FMD is relatively unknown in the medical practice, as the authors correctly state, and it would be useful to have a review of the literature on this technique.
However, there are specific concerns that need to be addressed:
- Language and flow need to be revised throughout the manuscript.
- Fig.1 is not really informative and it is unnecessary.
- The interpretation of results from literature and how these could affect the clinical practice are not sufficiently elaborated.
- It would be helpful to add a graphic representation of to have a visual snapshot of main conclusions.
Reviewer 2 Report
Reading this manuscript, this reviewer gets the impression that it is a student thesis. The authors have searched and read the literature carefully, but it seems evident they have not a reasonable experience (any at all ?) from measuring FMD. The text is lengthy, and not arranged as expected by Introduction and Conclusions. (Methods and Results may be combined as done). There are also many expressions as if this was written for a general population, not the medical profession (e.g. inner layer "called the intima", "endothelium is a......." . Why wording as "Most popular methods"). A Discussion is important as the final part.
Regarded a Student thesis my evaluation is that this work is admirably well done, but as a medical paper for the profession, it requires modifications.
Round 2
Reviewer 1 Report
The authors have been responsive, addressed this reviewer'scomments. The manuscript has improved.
Reviewer 2 Report
Modifications has improved the content